# Psychosocial Determinants of Stress Perceived among Polish Nursing Students during Their Education in Clinical Practice

**DOI:** 10.3390/ijerph19063410

**Published:** 2022-03-14

**Authors:** Iwona Bodys-Cupak, Lucyna Ścisło, Maria Kózka

**Affiliations:** 1Laboratory of Theory and Fundamentals of Nursing, Institute of Nursing and Midwifery, Faculty of Health Sciences, Jagiellonian University Medical College, 31-008 Krakow, Poland; 2Department of Clinical Nursing, Institute of Nursing and Midwifery, Faculty of Health Sciences, Jagiellonian University Medical College, 31-008 Krakow, Poland; lucyna.scislo@uj.edu.pl (L.Ś.); maria.kozka@uj.edu.pl (M.K.)

**Keywords:** stress, nursing, students, education, clinical practice

## Abstract

Background: Nursing students’ education process is related to the occurrence of difficult and stressful situations, especially during clinical placement. The purpose of the education is to develop critical thinking, clinical decision making and teamwork skills in students. This process should allow the students to integrate into the clinical environment and develop their professional identity. The goal of this research was to assess the relationship between perceived stress and psychosocial factors. Methods: The research was conducted in 2019 among 307 nursing students in Poland. Research questionnaires used in the study were: Perceived Stress Scale, Generalized Self-Efficacy Scale, Self-Esteem Scale, Life Satisfaction Scale, Life Orientation Test-R and Clinical Learning Environment Inventory. Results: There was a significant correlation between stress perceived by the surveyed nursing students and psychosocial components as well as teacher support and student’s satisfaction with clinical education. Satisfaction with the clinical education during the implementation of clinical activities was the highest in people experiencing a low level of stress. The highest level of teacher support was reported by people experiencing a higher level of stress. Conclusion: A higher level of perceived stress corresponded to a lower level of self-efficacy, lower life satisfaction, lower life orientation and lower self-esteem of students.

## 1. Introduction

One of the most crucial points of the education of nursing students is that the classes are conducted within a clinical environment, where the application of theoretical knowledge in practice is of key importance. The curriculum in a nursing school is aimed at preparing the graduates for working as independent practitioners and for applying the most up-to-date knowledge and skills in practice. The purpose of the education process is also to develop critical thinking, clinical decision making and teamwork skills in students. This process should allow the students to integrate into the clinical environment and develop their professional identity [1,2,3].

An inherent element in students’ education process is the occurrence of difficult and stressful situations. During undergraduate studies, nursing students in EU countries, including Poland, complete 2300 h of apprenticeship in direct contact with the patient. They are exposed to various stressors directly attributed to the clinical learning environment. In nursing school, especially during clinical classes, students experience new situations. They often perform complex activities that require exceptional manual dexterity. In addition, they come into contact with the pain and suffering of their patients, which causes stress. The perception of stress by the students in nursing schools is significantly higher as compared to the students of other disciplines [4,5,6,7,8,9].

The literature concerning stress distinguishes three ways of understanding the very idea of stress, namely: stress as a stimulus, stress as a response and stress as a transaction. These ways of understanding stress are not considered to be mutually exclusive but rather complementary [10,11,12].

Mildly intense stress may have a beneficial, motivating impact on the learning process, care and treatment measures or ensuring the safety of the patients. On the other hand, highly intense stress, especially long-term stress, has a negative impact on learning, as well as on achieving satisfactory learning outcomes, adaptation possibilities and students’ health and well-being. Excessive stress levels can have profound emotional, cognitive and physiological consequences [13,14,15].

Intense stress can manifest itself in the form of nausea and vomiting and irritability and may also have psychological consequences, such as depression, anxiety, low self-confidence, poor concentration and loss of motivation. Studies have also indicated that experiencing high levels of stress results in low academic success, lower levels of well-being, sleep disturbances and reduced quality of life [16,17,18].

These positive and negative effects of stress are inherently related to personal resources and, therefore, to the ability of students to cope with stress [19,20,21,22,23,24].

In the Polish literature, there are not many studies focused on the factors causing stress as perceived by the students of nursing during their clinical education, which was the reason for our decision to carry out this study. The authors hope that the results will allow for the verification of the existing studies and for presenting the problem in a different cultural context. The overall assessment of the factors causing the feeling of stress can help the students prepare for work in the often-burdensome environment of the health care system.

The goal of the research was to assess the relationship between perceived stress and a sense of optimism, a sense of self-efficacy, life satisfaction and self-esteem, as well as satisfaction with clinical activities and teacher support of Polish nursing students.

## 2. Materials and Methods

The study was conducted in 2019 among 307 people studying nursing at the level of first-cycle studies at the Jagiellonian University Collegium Medicum. Participants were purposefully selected for the study group. Sample size was calculated with G*Power 3.1.9.2, which revealed that a minimum of 207 participants was needed to perform a regression analysis. The following criteria for inclusion in the study were established: studying nursing in first-cycle studies, completion of practical classes in a hospital/clinic and informed consent to participate in the study. Students were excluded if they left one or more study semesters or were transferred from other nursing faculties during data collection (research duration). The participants were informed about the confidentiality and anonymity of the study, that participation in it was voluntary and they were allowed refuse to cooperate at any stage of the study.

Research team members distributed 450 questionnaires to students who met the study inclusion criteria. The tools were distributed during lectures, seminars and exercises. Study participants returned a total of 315 questionnaires. Ultimately, 307 questionnaires were used for the final analysis.

The research was a cross-sectional study, carried out by the diagnostic survey method, as well as through estimating and scaling. The research questionnaires used in the study were: Perceived Stress Scale (PSS10), Generalized Self-Efficacy Scale (GSES), Self-Esteem Scale (SES), Life Satisfaction Scale (SWLS), Life Orientation Test (LOT-R) and an inventory for the assessment of the clinical learning environment (CLEI-19) and a questionnaire developed by members of the research team, which included, among others, questions about sociodemographic variables. Some of the research tools were described in detail in the article “Psycho-social components determining the strategies of coping with stress in undergraduate Polish nursing students” published in BMC Nursing [19]. Below is just a general presentation of the research tools used.

The Perceived Stress Scale (PSS10) by S. Cohen, T. Kamarck and R. Mermelstein [21]. Z. Juczyński and N. Ogińska-Bulik adapted the tool to the Polish environment. It serves as a way to assess the intensity of stress related to one’s own life situation over the last month. The higher the respondent’s result, the greater the stress felt. The internal reliability of the original version of the scale, determined using the Cronbach’s alpha, ranges from 0.84 to 0.86 [12].

Generalized Self-Efficacy Scale (GSES) by R. Schwarzer and M. Jerusalem [25]. The tool was adapted to Polish conditions by Z. Juczyński. It measures the strength of an individual’s general belief in the effectiveness of coping with difficult situations and obstacles. The scale is characterized by a moderately high critical accuracy and reliability. Cronbach’s alpha coefficient for the scale is 0.85 [26].

Self-Esteem Scale (SES) by M. Rosenberg [27]. Adaptation to Polish conditions was carried out by M. Łaguna, K. Lachowicz-Tabaczek and I. Dzwonkowska. It is a 10-item self-report measure of global self-esteem. It consists of 10 statements related to overall feelings of self-worth or self-acceptance. The items are answered on a four-point scale ranging from strongly agree to strongly disagree. High internal consistency was demonstrated; Cronbach’s alpha coefficients for various age groups range from 0.81 to 0.83 [12,28].

Life Satisfaction Scale (SWLS) by E. Diener, RA Emmons, RJ Larson and S. Griffin [29]. Polish adaptation of the tool was carried out by Z. Juczyński. The scale assesses the general index of the person’s satisfaction with life. The higher the score, the higher the sense of their life satisfaction. Cronbach’s alpha coefficient for the scale is 0.85 [26].

Life Orientation Test (LOT-R) by MF Scheier, Ch. S. Carver and M. W. Bridges [30]. The adaptation was carried out by R. Poprawa and Z. Juczyński. The LOT-R tool consists of 10 theorems and enables the measurement of dispositional optimism. Six theorems (three positive and three negative) are used in the interpretation and calculation of the results. For the original version of the scale, the coefficient Cronbach’s alpha is 0.78 [26].

Inventory for the Assessment of the Clinical Teaching Environment (CLEI–19).

The CLEI-19 questionnaire by Y. Salamonson et al. [31] (2011) allows the assessment the student’s perceived support of the teacher in terms of learning and his (the student’s) satisfaction with the education in the hospital ward. The adaptation to Polish conditions was carried out by I. Bodys-Cupak [29]. The inventory contains 19 statements, with 12 relating to teacher support and 7 relating to satisfaction with the clinical education. In the Polish version of the Cronbach’s alpha scale, teacher support in regard to learning is 0.949 for the scale, whereas satisfaction with hospital stay is 0.901 for the scale. The values obtained indicate that the designed scale is a reliable tool [32].

The original questionnaire contained sociodemographic questions regarding, among others: gender, age and year of studies. It made it possible to collect information used for an in-depth characterization of the study group.

During the analysis of the collected research material, statistical methods were also used, allowing for the development of results and conclusions. The values of quantitative variables in two groups were compared using the Mann–Whitney test. The comparison of quantitative variable values in three or more groups was performed by means of the Kruskal–Wallis test. After detecting statistically significant differences, a post hoc analysis was performed with the Dunn’s test to identify statistically significantly different groups. Correlations between quantitative variables were analyzed using the Spearman’s or Pearson’s rank correlation coefficient. Additionally, regression analysis was performed using the input method. A significance level of 0.05 was assumed. The analysis was performed using the R program, version 3.6.1., Core Team (2019) [33].

### 2.1. Characteristics of the Study Group

The study group consisted of 307 people who met the participation criteria for the study; 96.1% were women (N = 295), and 3.9% were men (N = 12). The average age of the respondents was 20.82 ± 1.53, with their age ranging from 19 (N = 49) to 34 (N = 1). The average age of women was 20.78 ± 1.33, while the average age of men was 21.92 ± 4.06, and the 20- and 21-year-old groups were the biggest age groups (N = 90, i.e., 29.3%, and N = 86, i.e., 28.0%, respectively). A total of 20.2% of the participants were 22 years old (N = 62), and few were slightly older than 20 (N = 20, i.e., 6.5%). Freshmen constituted 40.1% of all the respondents (N = 123). A total of 37.5% of the participants were second-year students (N = 115), and 22.5% were third-year students (N = 69).

### 2.2. Ethical Considerations

The research was approved by the Bioethics Committee—No. of approval: 072.6120.208.2018

Students were informed of the confidentiality and anonymity of the study, that their participation was voluntary and that they were allowed to cease to cooperate at any time during the study.

The study was conducted in accordance with the principles of the Helsinki Declaration. All of the collected data are stored as protected files and accessible according to the regulations of the General Data Protection Regulation [34,35].

## 3. Results

The largest group among the respondents (N = 195, i.e., 63.5%) comprised students who perceived a high level of stress. The average level of stress was experienced by 28.7% of the respondents (N = 88), and the low level of stress by 7.8% (N = 24). The mean point score was 21.15 ± 5.30 points and ranged from 8 up to 36 points.

Students in their second year (32.2%) and third year (33.3%) perceived average levels of stress more frequently. A high level of stress was perceived by slightly more than 60% of students of every year. The differences were statistically significant (*p* = 0.0281); see Table 1.

Based on the GSES scale, it was found that more than half of the students (N = 179, i.e., 58.3%) had a high sense of self-efficacy. An average level was noted in 34.2% of the respondents (N = 105), and low self-efficacy was noted only in 7.5% of the respondents (N = 23). The average score on the GSES scale was 29.83 ± 4.24 points and ranged from 10 points to 40 points. More than half of the respondents (58%) had a high sense of self-efficacy, and 34.2% had an average sense of self-efficacy (Figure 1)

The majority of the students (N = 233, i.e., 75.9%) were characterized by a high level of self-esteem. The average SES score was 21.19 ± 4.58 points and ranged from 10 points to 32 points. The vast majority (75.9%) of the respondents were characterized by a high level of self-esteem, and 21.2% had an average score (Figure 2)

A low level of satisfaction with life was declared by 27.0% of students (N = 83). The average level of satisfaction with life was 20.82 ± 5.72 points and was in the range of 5–35 points. The average result on the sten scale was 5.64 ± 2.00 points and was in the range of 1–10 points.

Average results were obtained by 42.0% of the respondents and high results by 30.9% of the respondents (Figure 3).

A pessimistic attitude was presented by 30.0% of the students (N = 92). The average level of life orientation concerned 37.8% of the respondents (N = 116). In the case of 32.2% of people (N = 99), the life attitude was optimistic. The mean result of the LOT-R test was 14.43 ± 3.80 points, and the results were in the range of 2–24 points. (Figure 4)

On the basis of the CLEI-19 scale, it was shown that the average sum of teacher support in terms of learning was 31.73 ± 10.71 points and fluctuated in the range of 12–53 points. (The scale range is 12–60 points). Satisfaction with the education in the hospital ward, as a total result, amounted to an average of 26.22 ± 5.10 points and ranged from 11 to 35 points (the range is 7–35 points).

Taking into account the mean values of the two scales (1–5 point scale), it was possible to compare the results. It was thus found that satisfaction with the education in the hospital ward was higher (3.75 ± 0.73 points) than with the teacher’s support in learning (2.64 ± 0.89 points); see Table 2.

### 3.1. The Determinants of the Intensity of Stress Experienced by the Participants

The analysis has revealed statistically significant differences in the level of support and satisfaction with clinical education (taking into account the results of the PSS-10 scale in relation to the norms). The highest level of teacher support in terms of learning was reported by people experiencing average (33.42 points) or high (31.52 points) levels of stress. On the other hand, people experiencing low levels of stress reported lower teacher support (27.21 points). Satisfaction with the clinical education during the implementation of clinical activities was the highest in people experiencing a low level of stress (29.46 points), higher in people experiencing stress of moderate intensity (26.02 points) and the lowest in those experiencing stress of high intensity (25.91 points); see Table 3.

### 3.2. Correlation Analysis—Nonparametric Correlations (NONPAR CORR)

Correlation analysis indicated that higher levels of perceived stress corresponded to a lower level of self-efficacy, lower life satisfaction, lower life orientation and lower self-esteem. In the case of the self-esteem scale, rho > 0 indicates a lower self-esteem, as it is a negative scale. Therefore, it can be concluded that the lower the level of self-efficacy of the respondents, the lower their self-esteem, the lower their life satisfaction and the more pessimistic their attitude and the greater the intensity of the stress they felt (Table 4).

### 3.3. Regression Analysis by Input Method

The results of the regression analysis allowed for a conclusion that the intensity of stress perceived by the surveyed nursing students was higher in people who obtained a lower score for life satisfaction and in people who were more pessimistic about life (Table 5).

## 4. Discussion

Stress is present in the lives of students throughout the education process, especially during the clinical training. More than half of the students in the above study perceived stress of high intensity. Similar results were obtained by other researchers [5,8,36,37].

High levels of stress were perceived by nursing students in the last year of their studies in the studies carried out by Lavoie-Tremblay et al. (2021). This was attributed to specific clinical conditions that require the students to master complex technical skills and acquire extensive knowledge [38]. In our own research, a high level of stress was experienced by slightly more than 60% of students of each year.

In turn, the results of the studies carried out by Shaban et al. (2012) and Hamadi et al. (2021) allowed for a conclusion that most of the nursing students participating in the study experienced a moderate or low level of stress during clinical classes [39,40]. In our own research, students experienced a moderate level of stress, and 28.7% and 7.8% of the respondents experienced low level of stress.

The studies carried out by Bilgic and Celikkalp (2021) revealed that the gender of the students had no bearing on the level of perceived stress and their coping style [8]. Similar results were obtained in our own research. Moreover, it was found that higher levels of perceived stress were more often experienced by the students with lower self-esteem and by those with a lower sense of self-efficacy. The studies carried out by other authors have proven that stress has a negative impact on the sense of self-efficacy [41,42]. In addition, the analysis of the results of our own study conducted among students in Poland has revealed that a higher level of perceived stress was associated with lower life satisfaction and a more pessimistic attitude to life. Earlier studies (Yang and Kim, 2016, Labrague, 2021) also suggest that perceived stress was a significant predictor of a low level of satisfaction with life [43,44]. According to the results of the study by Rajan and Babu (2021), there was a significantly lower inverse correlation between the perceived stress and optimism among young adults, i.e., the higher the intensity of perceived stress, the more frequently they had a pessimistic attitude [45].

In addition, the results of studies carried out by Mousavi and Kamala (2021) among the students of nursing allowed for concluding that personality traits significantly correlated with the perceived level of stress within the clinical environment [46]. The most important predictors of stress in nursing students in a study by Admi et al. (2018) were the year of study and gender. The level of stress and satisfaction of second-year students before the start of the clinical classes were significantly higher compared to third and fourth year students. The female students experienced a much higher level of stress and satisfaction [7].

Our own study allowed for concluding that students most satisfied with the realization of clinical classes were those students of nursing who perceived a low level of stress. In the study by Moon and Jung (2020), similar conclusions were drawn that people experiencing lower levels of stress were more satisfied with clinical practice [47].

A negative learning environment is associated with the development of stress among students of nursing. The existence of such a relationship has been proven in the studies conducted by Jagoda and Rathnayaki (2021). The majority of the students surveyed by those researchers were moderately stressed and perceived the learning environment as “more positive than negative” [48].

## 5. Conclusions

The results of the conducted research have demonstrated that during the clinical training, nursing students perceived stress of moderate or high intensity. The level of stress was higher in people with low self-esteem who had a low sense of satisfaction with life and people who were more pessimistic about life. The results indicate the need to support the professional development of nursing students by increasing their self-esteem, sense of effectiveness and satisfaction with life, as well as professionalism.

Stress related to the clinical learning environment is a recurring problem in nursing education despite the increasing commitment and support provided to students. It is important and still relevant to detect the occurrence of this phenomenon and the factors influencing its severity in the best possible way. This will allow for the identification of needs and for the implementation of proper preventive measures, as well as for the reduction in the negative effects of stress. More research is also needed in that regard.

### Limitation of the Study

The main limitations of the study concern data collection methods. Data collection was carried out at a certain point in time, not longitudinally. The group of respondents involved women in particular, which may cause bias. Additionally, the research was carried out in a single university; therefore, the conclusions cannot be generalized. It is necessary to conduct multicenter studies to generalize the results and implement the recommendations.

## Figures and Tables

**Figure 1 ijerph-19-03410-f001:**
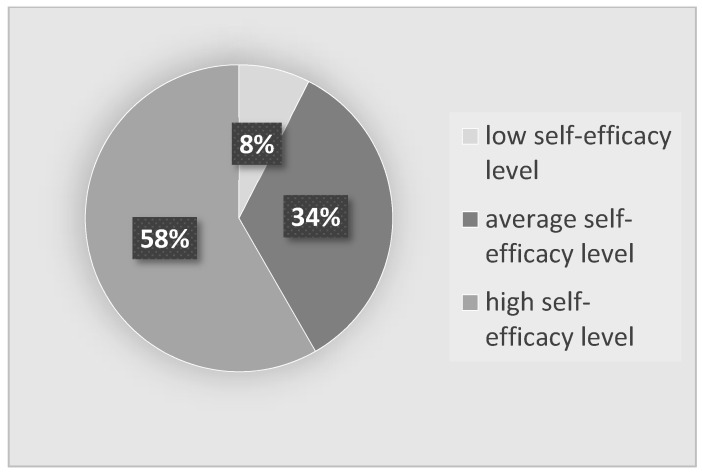
Self-efficacy of the nursing students participating in the study.

**Figure 2 ijerph-19-03410-f002:**
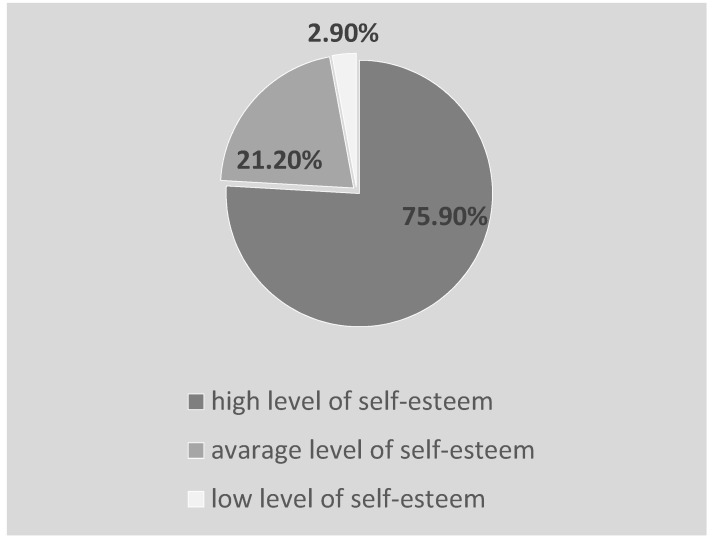
Self-esteem of the students participating in the study.

**Figure 3 ijerph-19-03410-f003:**
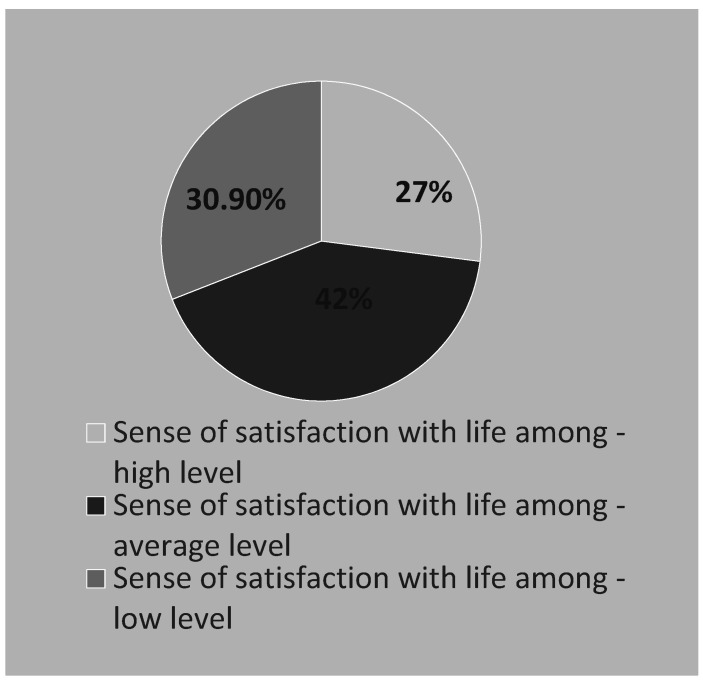
Sense of satisfaction with life among the respondents.

**Figure 4 ijerph-19-03410-f004:**
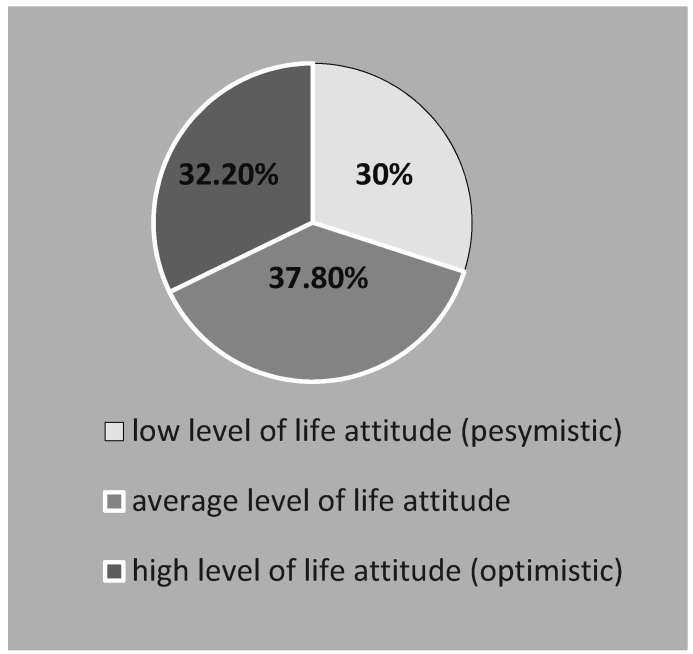
The life attitude of the surveyed students of nursing.

**Table 1 ijerph-19-03410-t001:** The level of intensity of perceived stress depending on the year of study.

Perceived Stress Scale (PSS-10)			Year of Study			Total
			1st Year	2nd Year	3rd Year	
The intensity of stress perceived by the students	Low intensity of perceived stress	N	16	7	1	24
		%	13.0%	6.1%	1.4%	7.8%
	Average level of perceived stress	N	28	37	23	88
		%	22.8%	32.2%	33.3%	28.7%
	High level of perceived stress	N	79	71	45	195
		%	64.2%	61.7%	65.2%	63.5%
*p*			0.0281			

*p* level of significance, n—number.

**Table 2 ijerph-19-03410-t002:** Support of the teachers and the level of participants’ satisfaction with the clinical education.

CLEI	Teacher Support in Learning (Total)	Satisfaction with Clinical Practice (Total)
M	31.73	26.22
Me	35.00	26.00
SD	10.71	5.10
Min.	12	11
Max.	53	35

M average, Me median, SD standard deviation, Min.—minimum, Max.—maximum.

**Table 3 ijerph-19-03410-t003:** The scale of perceived stress and the assessment of teacher support and satisfaction with clinical education.

Perceived Stress Scale (PSS-10)		Teacher Support in Learning (Total)	Satisfaction with Clinical Education (Total)
poor results	M	27.21	29.46
	SD	11.74	5.06
average results	M	33.42	26.02
	SD	10.50	4.89
good results	M	31.52	25.91
	SD	10.54	5.08
Total	M	31.73	26.22
	SD	10.71	5.10
*p*		0.0360	0.0085

*p* level of significance, M average, SD standard deviation.

**Table 4 ijerph-19-03410-t004:** Analysis of the correlation between the intensity of stress experienced by the participants and selected psychosocial variables.

			GSES (Total)	SWLS (Total)	SES (Total)	LOT-R (Total)	Support Total	Satisfaction Total
Spearman’s Rho	Perceived Stress Scale (PSS-10 total)	rho	−167	−0.376	0.214	−0.389	−0.026	−0.071
		*p*	0.0034	0.0000	0.0002	0.0000	0.649	0.212
		N	307	307	307	307	307	307

*p* level of significance, rho—Sperman correlation, N—total number of students.

**Table 5 ijerph-19-03410-t005:** The results of the regression analysis between the intensity of perceived stress and selected psychosocial variables.

Coefficients						
Regression Model		Non-Standardized Coefficients		Standardized Coefficients	*t*	*p*
		B	SE	Beta		
1	(Constant)	30.691	2.989		10.268	0.000
	Generalized Self-Efficacy Scale (GSES total)	0.012	0.073	0.009	0.159	0.874
	Satisfaction with Life Scale (SWLS sum)	−0.262	0.055	−0.283	−4.768	0.000
	Self-Esteem Scale (SES total)	0.052	0.065	0.045	0.808	0.419
	Life Orientation Test (LOT-R total)	−0.394	0.081	−0.283	−4.842	0.000
	Clinical Tutor Support	0.023	0.005	0.058	1.056	0.292
	Satisfaction with the clinical education			0.100	1.833	0.068

B nonstandardized regression coefficient, SE standard error, Beta standardized regression coefficient, *t* test *t*, *p* significance level  <  0.05.

## Data Availability

The data presented in this study are available on reasonable request from the corresponding author.

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
