# Peer review of "Psychosocial Determinants of Stress Perceived among Polish Nursing Students during Their Education in Clinical Practice"

_ijerph, 2022, doi:10.3390/ijerph19063410_

Round 1
Reviewer 1 Report
Although I am not a native I would suggest English proof-reading eg. skills in students (12), related the occurrence (10), and others also grammar issues (tenses for the sentence 81-82) justify this recommendation. Extensive editing is necessary in order to improve the paper (e.g. Line 109- text in Polish, figures too big do not suit to the rest of the text).
Well described methodology appropriate for the study. However some details should be improved – e.g. 307 vs 317 – number of students investigated lines 73 and 86 – not coherent
I do not understand why - The highest level of teacher support in terms of learning was 219 reported by people experiencing average (33.42 points) or high (31.52 points) level of 220 stress. It is important conclusion mentioned in the abstract.
The author promise to refer to cultural context (64-65) but I did not find it later on in the text.
Author Response
Dear Reviewer,
thank You very much for valuable comments.
Although I am not a native I would suggest English proof-reading eg. skills in students (12), related the occurrence (10), and others also grammar issues (tenses for the sentence 81-82) justify this recommendation. Extensive editing is necessary in order to improve the paper (e.g. Line 109- text in Polish, figures too big do not suit to the rest of the text).
Thank you for your valuable comments. Native speaker corrected the paper. The text was corrected, the content in Polish was removed. The figures have been adjusted in size to the text.
Well described methodology appropriate for the study. However some details should be improved – e.g. 307 vs 317 – number of students investigated lines 73 and 86 – not coherent
Thank you very much for this comment. 450 questionaires were distributed, students returned 315 completed questionnaires, which were checked, and 8 were rejected due to incomplete data. As a result, the analysis was conducted taking into account the results from 307 surveys.
I do not understand why - The highest level of teacher support in terms of learning was 219 reported by people experiencing average (33.42 points) or high (31.52 points) level of 220 stress. It is important conclusion mentioned in the abstract.
Thank you for this question. The authors' own research showed such a correlation that the high level of the teacher's support in the process of clinical education was indicated by students with a higher level of perceived stress. It may indicate that in the moment of experiencing a high or average stress, students find support in the teacher. It should be added that supporting students in the education process is the key in coping with stress.
The author promise to refer to cultural context (64-65) but I did not find it later on in the text.
The intention of the authors was to show the results in relation to other studies which are conducted among students studying in other countries, which was presented in the discussion. Therefore, the cultural theme was not developed. The results of the research on stress experienced by nursing students in the process of the clinical education are available mainly among authors from outside Poland.
Reviewer 2 Report
The present study focuses an important aspect of mental health of nursing students and the results could be useful to design a strategy to reduce the stress levels in these students. I have a few comments for the authors.
- For the following sentences, what do authors mean by tools? Is it the questionnaires? Tools appear to be a confusing term, can you replace it with some other word? The numbers also need to be checked once again as it looks to be a discrepancy. "Research team members distributed 450 questionnaires to students who met the study inclusion criteria. The tools were distributed during lectures, seminars and exercises. Study participants returned a total of 315 questionnaires. Ultimately, 317 pieces of the tool were used for the final analysis"
- Did authors tried to compare the stress levels of participants from different study years. This could help them to understand the cause of stress. Is it because of an entire new environment and later it reduces when the students spend a few years in this environment.
- There are almost more than 90% students with average or high level of stress however the number of average or high level of self esteem and self efficacy is also more than 90%. Even for the high level of stress, there are 65% respondents whereas low level of self esteem or self efficacy is reported by just 3% and 8% respectively. As per the conclusion, it appears stress and self efficacy/self esteem are positively correlated which is not true as per the table no. 5. Even for the high level of stress, there are 65% respondents whereas low level of self esteem or self efficacy is reported by just 3% and 8% respectively. Could you please explain this? Although, higher stress matches well with low optimism and life satisfaction.
- Second half of the first paragraph for the discussion is a bit confusing. Can authors rewrite it in a simple way? "In our own research a high level of stress was experienced by slightly more than 60% of students of each year. In turn, inThe results of the studies carried out by Shaban et al. (2012) and Hamadi et al. (2021) allowed for a conclusion that most of the nursing students participating in the study experienced moderate or low level of stress during clinical classes [42,43]. In our own research the numbers were 28.7% and 7.8% of the respondents, respectively."
- There are several spelling errors in the draft, I would suggest authors to check the grammar once again. Use of "." or "," should be consistent for decimal.
Author Response
Dear Reviewer,
Thank you very much for your valuable comments.
For the following sentences, what do authors mean by tools? Is it the questionnaires? Tools appear to be a confusing term, can you replace it with some other word? The numbers also need to be checked once again as it looks to be a discrepancy. "Research team members distributed 450 questionnaires to students who met the study inclusion criteria. The tools were distributed during lectures, seminars and exercises. Study participants returned a total of 315 questionnaires. Ultimately, 317 pieces of the tool were used for the final analysis"
Thank you very much for this comment. Tools means questionaires. 450 questionaires were distributed, students returned 315 completed questionnaires, which were checked, and 8 were rejected due to incomplete data. As a result, the analysis was conducted taking into account the results from 307 surveys.
Did authors tried to compare the stress levels of participants from different study years. This could help them to understand the cause of stress. Is it because of an entire new environment and later it reduces when the students spend a few years in this environment.
In this study, the results allowed to conclude that the highest level of perceived stress was reported by the first-year students, while the students of the second and third years perceived stress of an average intensity. Nevertheless, as many as 60% of the students from each year indicated that they felt stress of a high intensity. However, the same students were not studied, which may have an impact on the results and is one of the limitations of the research. Thank you very much for this inspiring attention, we will use the idea when planning the next research.
There are almost more than 90% students with average or high level of stress however the number of average or high level of self esteem and self efficacy is also more than 90%. Even for the high level of stress, there are 65% respondents whereas low level of self esteem or self efficacy is reported by just 3% and 8% respectively. As per the conclusion, it appears stress and self efficacy/self esteem are positively correlated which is not true as per the table no. 5. Even for the high level of stress, there are 65% respondents whereas low level of self esteem or self efficacy is reported by just 3% and 8% respectively. Could you please explain this? Although, higher stress matches well with low optimism and life satisfaction.
The analysis of the research's results was conducted in several stages. Firstly, the categories (low/average/high) were created for the scales used in the study and the chi-square test of independence was used to evaluate the relationship between the variables. However, the use of the correlation is more reliable, therefore the Sperman rank correlation has been calculated, because it reflects the actual state more closely. On the other hand, as indicated in the review, the relationship between the level of stress and the results of the SWLS and LOT-R scales is clearly visible, the correlation with the sense of the effectiveness or self-esteem is weak (the rho values lower than 0.3 are interpreted as a weak correlation). The confirmation of a stronger relationship was performed in the next stage of the step-linear regression analyses, which indicated that only the SWSL and LOT-R significantly influenced PSS-10; the influence of GSES and SES was not significant and it was directly manifested after dividing the results into low/medium/high and performing the chi-square test of independence.
Second half of the first paragraph for the discussion is a bit confusing. Can authors rewrite it in a simple way? "In our own research a high level of stress was experienced by slightly more than 60% of students of each year. In turn, inThe results of the studies carried out by Shaban et al. (2012) and Hamadi et al. (2021) allowed for a conclusion that most of the nursing students participating in the study experienced moderate or low level of stress during clinical classes [42,43]. In our own research the numbers were 28.7% and 7.8% of the respondents, respectively."
Thank you very much for that comment. The text has been simplified and redrafted.
There are several spelling errors in the draft, I would suggest authors to check the grammar once again. Use of "." or "," should be consistent for decimal.
Thank you very much for that valuable comment. The marking of decimals was standardized.